# Measurement of Limbic System Anatomical Volumes in Patients Diagnosed with Schizophrenia Using Vol2brain and Comparison with Healthy Individuals

**DOI:** 10.3390/medicina61030525

**Published:** 2025-03-17

**Authors:** Mert Ocak, Buket Oguz

**Affiliations:** 1Department of Basic Medical Science—Anatomy, Faculty of Dentistry, Ankara University, 06100 Ankara, Turkey; 2Department of Anatomy, Faculty of Medicine, Ankara Yildirim Beyazit University, 06760 Ankara, Turkey; buketoguzbeyoglu@gmail.com

**Keywords:** schizophrenia, limbic system anatomy, cortical thickness, amygdala volume, hippocampal morphology, cingulate gyrus structure

## Abstract

*Background and Objectives*: Schizophrenia is a chronic psychiatric disorder affecting approximately 24 million people worldwide, characterized by structural and functional brain abnormalities. Despite its prevalence, automated segmentation tools like Vol2Brain have been underutilized in large-sample studies examining limbic system anatomical volumes in patients with schizophrenia. This study aimed to assess volume differences in all major limbic system structures between schizophrenia patients and healthy controls using Vol2Brain. *Method*: This retrospective study included 68 schizophrenia patients and 68 healthy controls, with MRI scans obtained from OpenNeuro. Limbic system volumetric and cortical thickness measurements were conducted using Vol2Brain, an automated segmentation platform. *Results*: Schizophrenia patients exhibited significantly reduced volumes in the amygdala, hippocampus, anterior cingulate gyrus, posterior cingulate gyrus, and middle cingulate gyrus compared to controls. However, the left amygdala volume was larger in schizophrenia patients. A cortical thickness analysis revealed that schizophrenia patients had thinner limbic cortices, particularly in the anterior and posterior cingulate gyri and the right parahippocampal gyrus. In contrast, the right anterior cingulate gyrus was thicker in schizophrenia patients. The differences in total and left parahippocampal gyrus volumes and cortical thickness did not reach statistical significance. *Conclusions*: These findings reinforce previous evidence of limbic system abnormalities in patients with schizophrenia, which may contribute to cognitive and emotional dysregulation. The study also highlights Vol2Brain’s potential as a rapid, cost-free, and reliable alternative for brain volume analysis, facilitating more standardized and reproducible neuroimaging assessments in psychiatric research.

## 1. Introduction

Schizophrenia is a chronic psychiatric disorder affecting approximately 24 million people worldwide, characterized by a distorted perception of reality. It manifests through symptoms such as hallucinations, anhedonia, and motor dysfunction, which are associated with functional and structural changes in the brain [1,2]. Various brain regions, including the frontal gyri, thalamus, temporal gyri, and limbic areas, have been implicated in the pathophysiology of schizophrenia [3].

Magnetic resonance imaging (MRI) studies have identified significant volumetric alterations in several anatomical regions of the brain in individuals with schizophrenia [4,5]. Among these regions, the limbic system plays a critical role due to its involvement in emotional regulation, behavioral responses, memory, and learning. The limbic system includes key structures such as the parahippocampal gyrus, hippocampus, and cingulate gyrus [6].

Vol2Brain is an advanced, free, and online MRI volumetric application (accessible at https://volbrain.net/ (accessed on 1 December 2024) designed for the automated segmentation of brain structures [7]. This tool offers rapid and accurate volumetric calculations, significantly reducing error rates compared to manual segmentation methods. By leveraging technologies such as multi-atlas labeling and deep learning, Vol2Brain enhances the diagnostic process for neurological disorders [8].

Despite its potential, a review of the existing literature highlights a gap in large-sample studies specifically analyzing limbic system structures in schizophrenia patients compared to healthy controls using automated segmentation tools like Vol2Brain. Many existing studies utilizing Vol2Brain are limited by small sample sizes or focus on schizophrenia in the context of comorbid conditions, such as bipolar disorder [9,10].

The primary aim of this study was to assess the anatomical volumes of limbic system components in adults who were diagnosed with schizophrenia and to investigate potential differences in limbic system volume loss compared to healthy individuals in a large sample. Additionally, this study seeks to promote the widespread adoption of Vol2Brain as a reliable, user-friendly, and cost-effective alternative to traditional manual segmentation techniques for brain volume analysis.

## 2. Materials and Methods

### 2.1. Subjects and Sample

This retrospective study included 68 individuals who were diagnosed with schizophrenia and 68 healthy controls, resulting in a total of 136 adult participants. Magnetic resonance (MR) images of these participants were retrieved from OpenNeuro [(RRID:SCR_005031), accessible at https://openneuro.org (accessed on 1 December 2024)], a free and open-source data repository hosting various types of brain imaging data. OpenNeuro facilitates data sharing in compliance with the FAIR principles, which aim to enhance the findability, accessibility, interoperability, and reusability of research data, thus improving reproducibility in scientific studies [11].

Participants aged 18 years and older were included in the study and categorized into two groups: the group diagnosed with schizophrenia and the healthy control group. The participants were adults aged 18 years and older, as OpenNeuro datasets primarily include data from adult populations. However, specific demographic details such as the exact age distribution and gender were not provided in the dataset used in this study. The schizophrenia group consisted of individuals who had received a confirmed clinical diagnosis of schizophrenia, while the control group included individuals with no history of neurological or psychiatric disorders. The exclusion criteria for the schizophrenia group included comorbid neuropsychiatric conditions, significant structural brain abnormalities unrelated to schizophrenia, or a history of traumatic brain injury.

The dataset was retrieved from OpenNeuro, which provides open access data under Creative Commons CC0 and ensures compliance with ethical and legal standards for data sharing. The dataset is anonymized, and the original study obtained informed consent from all participants for data collection and sharing in a de-identified form, as stated in the repository’s documentation.

### 2.2. Structural MR Data Analysis

Following registration with the OpenNeuro database, MR imaging data for both the schizophrenia and control groups were downloaded within one week. Tissue segmentation, structural segmentation, and volumetric measurements of the MR images were conducted using the Vol2Brain platform, producing detailed reports for each participant (Figure 1). The automated segmentation and volumetric analysis provided by Vol2Brain ensure a high degree of accuracy and reproducibility, making it a reliable tool for neuroimaging research.

The anatomical regions of the limbic system that were analyzed in this study included the total limbic system, amygdala, hippocampus, anterior cingulate gyrus, middle cingulate gyrus, posterior cingulate gyrus, and parahippocampal gyrus. For each of these structures, volumetric measurements were performed in cubic centimeters (cm^3^) for the total volume, as well as for the right and left sides separately.

Additionally, the gray matter thicknesses of the total limbic cortex, anterior cingulate gyrus, middle cingulate gyrus, posterior cingulate gyrus, and parahippocampal gyrus were calculated in millimeters (mm) for the total and the right and left sides. A cortical thickness mapping of the analyzed structures was generated, providing a visual representation of the measurements (Figure 2).

### 2.3. Statistical Analysis

Quantitative data obtained in this study were subjected to normality tests before proceeding with statistical analysis. The Shapiro–Wilk test (normal distribution assumed if *p* > 0.05), Q-Q plots, and histograms were used to assess the normality of the data distribution.

For data following a normal distribution, comparisons of means were conducted using the Independent T-test. For non-normally distributed data, the Mann–Whitney U test was applied. Differences in structure volumes between schizophrenia patients and healthy controls were considered statistically significant at *p* < 0.05.

All quantitative data analyses were performed using IBM SPSS Statistics version 26. Ethical approval for the study was granted by the Ethics Committee of Ankara Yildirim Beyazit University (2024-1027).

## 3. Results

The mean volumes and cortical thicknesses of the limbic system components, as measured by Vol2Brain, are presented in tables, comparing healthy individuals and patients diagnosed with schizophrenia.

### 3.1. Comparison of Volumes

The total and right amygdala volumes were larger in healthy individuals, whereas the left amygdala volume was significantly larger in patients who were diagnosed with schizophrenia. In addition, the volumes of the hippocampus, anterior cingulate gyrus, posterior cingulate gyrus, and the entire limbic system were larger in healthy individuals compared to patients with schizophrenia.

The total volume of the middle cingulate gyrus, as well as the right and left middle cingulate gyrus volumes, were larger in healthy individuals; however, these differences did not reach statistical significance. Similarly, the total parahippocampal gyrus volume was larger in healthy individuals, but the difference was not statistically significant. Notably, the right parahippocampal gyrus volume was significantly larger in healthy individuals, whereas the left parahippocampal gyrus volume was larger in patients with schizophrenia, although this difference was not statistically significant (Table 1).

### 3.2. Comparison of Cortical Thickness

The total, right, and left limbic cortical thicknesses were greater in healthy individuals; however, these differences were not statistically significant. In patients who were diagnosed with schizophrenia, the total thickness of the anterior cingulate gyrus and the thickness of the right anterior cingulate gyrus were larger. Although the thickness of the left anterior cingulate gyrus was also greater in schizophrenia patients, this difference was not statistically significant.

The total thickness of the middle cingulate gyrus, as well as the right and left middle cingulate gyrus thicknesses, were similar in both groups, with healthy individuals showing slightly thicker cortices, although the differences were not statistically significant.

Significant differences were observed in the posterior cingulate gyrus, where the total, right, and left cortical thicknesses were notably larger in healthy individuals. For the parahippocampal gyrus, the total and left cortical thicknesses were greater in healthy individuals, but these differences were not statistically significant. However, the thickness of the right parahippocampal gyrus was significantly greater in healthy individuals compared to patients with schizophrenia (Table 2).

## 4. Discussion

This study constitutes a significant piece of research, the aim of which was to analyze the structural changes that occur in the limbic system components of patients who are diagnosed with schizophrenia. In comparison with previous studies, a larger sample group was utilized, and all major components of the limbic system were analyzed in detail. Moreover, this study employed automated segmentation techniques, ensuring methodological consistency and facilitating a more precise assessment of changes in the limbic system.

As demonstrated in Sigmundsson et al. [12], previous studies have indicated that schizophrenia is predominantly associated with structural changes in the frontal, temporal, and limbic regions. It is recognized that components of the limbic system are intimately linked to emotional processing, memory, and cognitive control mechanisms in individuals who are diagnosed with schizophrenia [13]. In this context, the alterations in limbic system volumes and cortical thicknesses that were observed in this study offer significant insights into the neuropathology of the disease.

### 4.1. Volumes

The amygdala, a component of the limbic system, has been identified as playing a crucial role in emotional processing and threat perception mechanisms [14]. In accordance with the current literature, our study found that the total and right amygdala volumes were significantly diminished in individuals who were diagnosed with schizophrenia [9,15,16]. This finding lends further support to the hypothesis that decreases in amygdala volume may be associated with negative symptoms (e.g., social withdrawal, emotional blunting) that are commonly observed in individuals who are diagnosed with schizophrenia. In contrast to these findings, an increase in the left amygdala volume was observed. Previous studies have reported no significant reduction in amygdala volume in postmortem analyses [17]. These inconsistencies imply that alterations in the amygdala volume in patients with schizophrenia may vary depending on disease subtypes, symptom profiles, or responses to treatment. Specifically, functional MRI studies demonstrate that patients with schizophrenia exhibit abnormalities in amygdala activity, which may be associated with psychotic symptoms [18]. Consequently, the evaluation of changes in amygdala volume as a biomarker in the diagnosis and prognosis of schizophrenia is recommended.

The hippocampus, a critical limbic structure that is implicated in memory and spatial learning processes, exhibits frequent structural alterations in individuals who are diagnosed with schizophrenia [19]. The reduction in total hippocampal volume, as well as right and left hippocampal volumes in schizophrenia patients, aligns with findings in the literature [20]. This supports the hypothesis that a hippocampal volume reduction could serve as a biomarker for diagnosis and disease monitoring [21]. These findings are particularly significant for their potential clinical contributions to the early detection and ongoing monitoring of schizophrenia. Notably, the observation of significant decreases in hippocampus volume during the early stages of schizophrenia suggests that this region may serve as a vital indicator in the early diagnosis of the disease

The anterior cingulate gyrus plays an important role in cognitive control and emotional regulation [22]. Consistent with a meta-analysis, the total, right, and left anterior cingulate gyrus volumes were reduced in schizophrenia patients [23]. The heterogeneity observed in previous studies, as noted in the meta-analysis, highlights the need for standardized methodologies. By employing an automated segmentation tool such as Vol2Brain, our study provides a roadmap for standardizing anterior cingulate gyrus measurements, which could be critical in considering this region as a therapeutic target.

Similarly, the total, right, and left posterior cingulate gyrus volumes were significantly reduced in schizophrenia patients, which is consistent with prior research [24,25]. These findings underline the potential of posterior cingulate gyrus volume as a significant biomarker for schizophrenia. The posterior cingulate gyrus is a region associated with verbal and visual memory processes and may be directly related to the cognitive impairments that are observed in patients with schizophrenia [26]. Incorporating posterior cingulate gyrus volume measurements into clinical practice could aid in tracking disease progression. Our study demonstrates that automated tools like Vol2Brain are reliable and efficient for this purpose.

In this study, the total, right, and left middle cingulate gyrus volumes were numerically lower in schizophrenia patients, although the difference was not statistically significant. The middle cingulate gyrus is a key structure associated with cognitive control and emotional balance and may be altered in patients with schizophrenia [27]. Previous studies with smaller sample sizes have reported similar findings, noting a reduction in the volume of the middle cingulate gyrus, but the variability in study results highlights the need for more standardized methodologies and larger cohorts [9,28]. Given that the cingulate cortex is highly connected to prefrontal and limbic structures, abnormalities in the middle cingulate gyrus may result in deficits in executive processing, motivation, and emotional processing [29]. In addition, some neuroimaging studies reveal that impairments in the middle cingulate region cause cognitive inflexibility and negative symptoms in patients with schizophrenia [30]. Limited research exists on the middle cingulate gyrus in patients with schizophrenia, and future studies with larger samples are needed to explore this relationship further. Our study, with a larger sample size than prior studies, serves as a foundation for such research. The abnormalities of the middle cingulate gyrus are linked to persistent negative symptoms in patients with schizophrenia, as are abnormalities in the posterior cingulate gyrus [31,32,33]. By employing a larger sample size and an automatic segmentation method, our study differs from earlier research and provides a more robust analysis.

Finally, the right parahippocampal gyrus volume was significantly reduced in schizophrenia patients compared to healthy controls, while the total and left parahippocampal gyrus volumes showed no significant differences. This aligns with a 2005 study that found no significant changes in parahippocampal gyrus volumes but noted a larger left parahippocampal gyrus volume compared to the right [34]. Additionally, a meta-analysis identified the right parahippocampal gyrus volume as an asymmetric region in patients with schizophrenia [35]. By employing Vol2Brain and adopting a novel methodological approach, our study contributes valuable insights to the ongoing investigation of parahippocampal gyrus volume in patients with schizophrenia.

### 4.2. Cortical Thickness

Changes in cortical thickness in patients with schizophrenia provide important information about the symptom profile and neuropathological mechanisms of the disease [36]. The relationship between cortical thickness and psychotic symptoms in patients with schizophrenia has been well established [37]. In this study, we aimed to assess the cortical thickness of limbic system components using a fast and highly accurate automatic segmentation method. The total and right anterior cingulate gyrus thicknesses were found to be reduced in schizophrenia patients. Previous studies support this finding, indicating that the anterior cingulate gyrus is generally thinner in patients with schizophrenia as part of the broader pattern of gray matter abnormalities that has been observed in these patients [38]. Considering the function of the anterior cingulate gyrus in cognitive flexibility and error detection processes, it is hypothesized that the observed reduction in thickness in this region may represent a primary mechanism underlying the cognitive impairment associated with schizophrenia [39].

Similarly, the total, right and left posterior cingulate gyrus thicknesses were thinner in schizophrenia patients in our study. However, some studies propose that the posterior cingulate gyrus thickness may be increased as a compensatory mechanism to enhance potential robustness in schizophrenia patients [40]. This discrepancy underscores the need for future studies to investigate the posterior cingulate gyrus thickness in patients with schizophrenia more comprehensively.

The parahippocampal gyrus is an essential part of the limbic system, because it is involved in important tasks such as memory processing, emotional stability, and spatial navigation. Abnormalities in this region can lead to cognitive impairment, memory lapses, and emotional deviations [41,42]. In our study, the right parahippocampal gyrus thickness was also thinner in schizophrenia patients. This finding aligns with previous research demonstrating bilateral atrophy of the parahippocampal gyrus thickness in patients with schizophrenia [43]. In previous studies using MRI and voxel-based morphometry, reduced parahippocampal thickness has been associated with reduced reality perception and delusional thoughts. The disintegration of the structural integrity of the parahippocampal gyrus may weaken memory processing so that hallucinations and delusions may occur. Measuring the parahippocampal thickness may contribute to the early diagnosis of schizophrenia and more timely intervention strategies that can help reduce the severity of the illness [44,45]. Furthermore, monitoring changes in this region may provide valuable information about the effectiveness of different approaches to treatment, such as cognitive behavioral therapy and neurostimulation techniques and may provide a personalized approach to patient management.

The cortical thicknesses of other limbic system components, including the total limbic cortex, middle cingulate cortex, and total and left parahippocampal gyrus, were similar across the entire sample. Although these measures were thinner in schizophrenia patients, the differences were not statistically significant. Notably, the left anterior cingulate cortex thickness was greater in schizophrenia patients compared to other cortical regions, although this difference was also not statistically significant. Previous studies suggest that the increased thickness in the left anterior cingulate gyrus in schizophrenia patients may be linked to childhood trauma or elevated schizotypy [46]. This finding suggests a potential link between increased anterior cingulate cortex thickness in patients with schizophrenia and early-life adversity. Consequently, targeted interventions such as trauma-focused cognitive behavioral therapy or mindfulness-based stress reduction may prove efficacious in mitigating associated cognitive and emotional dysfunctions. Further research is necessary to ascertain whether these thickness differences persist over time or fluctuate based on the symptom severity, treatment response, or environmental factors.

Future studies with larger sample sizes are needed to explore the potential reasons for the increased anterior cingulate cortex thickness in the left hemisphere of schizophrenia patients. These investigations could provide further insights into the role of cortical thickness variations in the pathophysiology and clinical manifestations of schizophrenia.

### 4.3. Vol2Brain and Alternative Segmentation Methods

In the context of the secondary objective of the present study, namely the promotion of Vol2Brain as an alternative to conventional manual segmentation techniques, a comparison of Vol2Brain with other widely utilized automated neuroimaging tools, including FreeSurfer v7.4.1., FMRIB Software Library, and Advanced Normalization Tools, is necessary.

Vol2Brain is a cloud-based, automated platform that provides fast and accurate brain volumetric analysis. Compared to the highly accurate but computationally demanding FreeSurfer, Vol2Brain offers faster processing and a user-friendly workflow [47]. Similarly, while FMRIB Software Library provides powerful structural segmentation, it requires complex pre-processing, making it less accessible to non-experts [48]. Advanced Normalization Tools, recognized for its high-precision multi-atlas registration, is computationally intensive and requires substantial expertise [49]. In contrast, Vol2Brain’s deep learning-based segmentation is fully automated, rendering it ideal for large-scale clinical and research applications.

### 4.4. Capability of Vol2Brain in Schizophrenia

The distinguishing feature of Vol2Brain is that the user bias that occurs in manual and semi-automatic segmentation is minimized. Manual segmentation has been shown to cause personal variability in the identification of complex structures such as the limbic system [50]. Since this process is carried out automatically when using Vol2Brain, a repeatable and standardized level of comfort of use is ensured.

A limitation of Vol2Brain is that it does not always generalize well, because it is dependent on a previously trained deep learning model. Although Free Surfer has been validated in a large sample of schizophrenia patients, the use of Vol2Brain for this condition is more recent [47]. Therefore, in future research, comparing manual segmentation and Vol2Brain using different patient populations may help to focus on accurate results.

### 4.5. Limitations and Future Directions

One limitation of this study is the exclusive use of the OpenNeuro database, which may lack sufficient diversity in its sample population. The generality of our findings is therefore limited, as factors such as genetic background, sex, age, and cultural variations may play a role in the observed structural differences in patients with schizophrenia. Future studies incorporating multi-center approaches or utilizing larger, more diverse databases would enhance the generalizability and validity of the findings.

Additionally, this study includes results that were not statistically significant, particularly in the middle cingulate gyrus and parahippocampal thickness measurements. Although these findings provide valuable insights into potential structural differences in patients with schizophrenia, their clinical relevance remains uncertain. The use of longitudinal study designs with more diverse participant groups including prodromal, first-episode, and chronic phases could provide a clearer clinical interpretation of limbic system volumes in patients with schizophrenia, a disease that is characterized by heterogeneous symptoms.

Further investigations could also focus on comparing the volumes and cortical thicknesses of limbic system components across different stages or subtypes of schizophrenia. In particular, the anterior and posterior cingulate gyrus warrant detailed exploration in this context. By examining the relationship between the structural characteristics of these regions and clinical symptoms, future research may help identify novel therapeutic targets for managing the disease.

The extensive utilization of automated segmentation tools such as Vol2Brain carries significant implications for both clinical and research contexts. Clinically, the expeditious and dependable analyses enabled by Vol2Brain facilitate the early diagnosis of patients. Moreover, within the realm of research, the integration of Vol2Brain with techniques such as functional MRI or Diffusion Tensor Imaging illuminates the correlation between the structures of interest and functions in patients with schizophrenia [51]. In the future, studies aiming to optimize the calculation of brain volumes to improve the precision of machine learning and individualized treatment methods can be carried out.

In spite of the absence of individual demographic details in the OpenNeuro dataset, the inclusion of a well-balanced sample size between the schizophrenia and control groups strengthens the validity of our findings. It is recommended that future studies utilize novel datasets encompassing comprehensive demographic characteristics, thereby facilitating more profound subgroup analyses based on variables such as age, sex, and illness duration.

## 5. Conclusions

In this study, the volumes and cortical thicknesses of limbic system anatomical components were compared between schizophrenia patients and healthy controls using the Vol2Brain automatic segmentation method. The volume reductions that were observed in regions such as the amygdala and hippocampus are believed to have significant potential in contributing to the development of biomarkers for schizophrenia, with implications for both the existing literature and clinical practice.

Our findings indicate that the volumetric and cortical thicknesses changes in the limbic system anatomy may serve as valuable tools for disease diagnosis and monitoring. Furthermore, we anticipate that the adoption of rapid, cost-free, and highly accurate automatic segmentation methods like Vol2Brain will expand, driving further advancements in neuroimaging research and clinical applications.

## Figures and Tables

**Figure 1 medicina-61-00525-f001:**
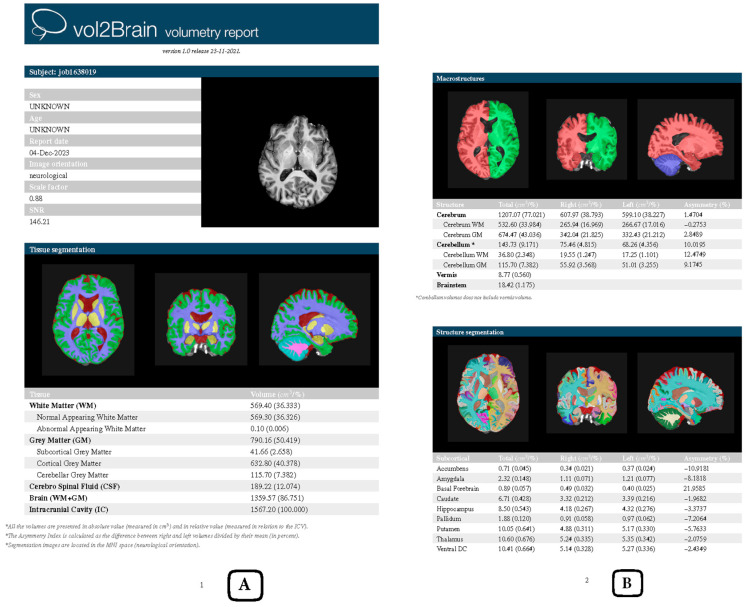
Example of a Vol2Brain volumetry report. The report includes anonymized patient details such as job ID, sex, age, and imaging parameters. It presents tissue segmentation results for WM, GM, CSF, and IC, with volumes in cubic centimeters and percentages (**A**). The macrostructure section details the cerebrum, cerebellum, vermis, and brainstem, while the subcortical segmentation highlights regions like the amygdala, hippocampus, and thalamus, including asymmetry indices (**B**). Axial, sagittal, and coronal views visually confirm the segmentation quality. Source: own study.

**Figure 2 medicina-61-00525-f002:**
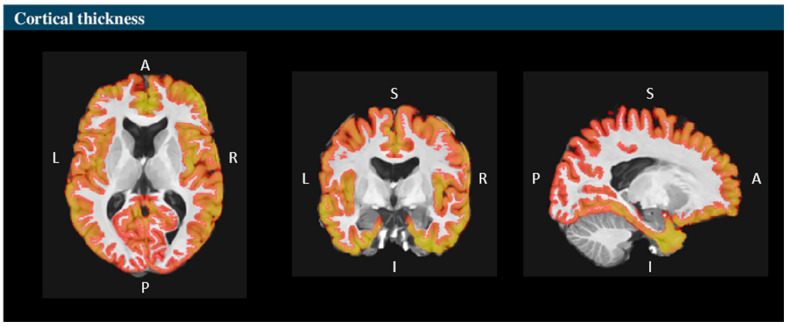
Cortical thickness mapping based on Vol2Brain segmentation. This figure displays cortical thickness measurements in the axial, coronal, and sagittal views, with a color-coded representation indicating regional variations. Warmer colors correspond to thicker cortical areas, while cooler tones indicate thinner regions. The segmentation process ensures precise anatomical delineation, highlighting critical brain structures. This visualization improves clarity by directly showing the cortical regions that were analyzed in the study. Source: own study.

**Table 1 medicina-61-00525-t001:** The mean volumes of the limbic system components in patients diagnosed with schizophrenia and healthy controls (cm^3^).

	Patients Diagnosed with Schizophrenia (*n* = 68)	Healthy Controls (*n* = 68)	*p*-Value	t-Value	z-Value
Total amygdala volume	2.07 ± 0.24	2.25 ± 0.25	0.001	−4.202	
Right amygdala volume	1.03 ± 0.12	1.12 ± 0.12	0.001		−3.652
Left amygdala volume	1.61 ± 0.20	1.13 ± 0.13	0.006		−2.725
Total hippocampus volume	8.08 ± 1.06	8.68 ± 0.96	0.001		−3.547
Right hippocampus volume	4.11 ± 0.56	4.37 ± 0.46	0.003		−3.003
Left hippocampus volume	4.01 ± 0.44	4.31 ± 0.38	0.001	−4.264	
Total limbic cortex volume	42.83 ± 7.48	47.77 ± 4.18	0.001		−5.075
Right limbic cortex volume	21.49 ± 3.11	23.37 ± 2.55	0.001		−4.100
Left limbic cortex volume	22.02 ± 3.05	24.39 ± 2.02	0.001		−5.188
Total anterior cingulate gyrus volume	12.32 ± 2.01	13.91 ± 1.91	0.001		−4.472
Right anterior cingulate gyrus volume	5.98 ± 1.36	6.58 ± 1.25	0.014		−2.468
Left anterior cingulate gyrus volume	6.27 ± 1.05	7.31 ± 1.10	0.001	−5.590	
Total middle cingulate gyrus volume	10.79 ± 1.67	11.85 ± 1.78	0.001		−4.596
Right middle cingulate gyrus volume	5.40 ± 0.93	6.00 ± 0.67	0.001		−4.274
Left middle cingulate gyrus volume	5.39 ± 0.87	5.99 ± 0.73	0.001	−4.392	
Total posterior cingulate gyrus volume	9.24 ± 1.69	10.61 ± 1.10	0.001		−5.680
Right posterior cingulate gyrus volume	4.65 ± 0.66	5.10 ± 0.68	0.001		−3.205
Left posterior cingulate gyrus volume	4.66 ± 0.92	5.51 ± 0.64	0.001		−5.915
Total parahippocampal gyrus volume	6.57 ± 0.73	6.72 ± 0.51	0.146 (*p* > 0.05)		
Right parahippocampal gyrus volume	3.17 ± 0.36	3.34 ± 0.29	0.003	−3.074	
Left parahippocampal gyrus volume	3.39 ± 0.42	3.38 ± 0.28	0.8 (*p* > 0.05)		

**Table 2 medicina-61-00525-t002:** The mean cortical thicknesses of the limbic system components in healthy individuals and patients diagnosed with schizophrenia (mm).

	Patients Diagnosed with Schizophrenia (*n* = 68)	Healthy Controls (*n* = 68)	*p*-Value	t-Value	z-Value
Limbik cortex total thickness	3.44 ± 0.43	3.50 ± 0.19	0.7 (*p* > 0.05)		
Right limbic cortex thickness	3.49 ± 0.22	3.50 ± 0.19	0.9 (*p* > 0.05)		
Left limbic cortex thickness	3.47 ± 0.23	3.50 ± 0.20	0.4 (*p* > 0.05)		
Anterior cingulate gyrus total thickness	4.11 ± 0.22	4.00 ± 0.24	0.005		−2.785
Right anterior cingulate gyrus thickness	4.13 ± 0.22	3.97 ± 0.24	0.001		−3.969
Left anterior cingulate gyrus thickness	4.10 ± 0.26	4.03 ± 0.27	0.14 (*p* > 0.05)		
Middle cingulate gyrus total thickness	3.30 ± 0.35	3.32 ± 0.26	0.9 (*p* > 0.05)		
Right middle cingulate gyrus thickness	3.31 ± 0.37	3.32 ± 0.29	0.7 (*p* > 0.05)		
Left middle cingulate gyrus thickness	3.28 ± 0.36	3.30 ± 0.25	0.8 (*p* > 0.05)		
Posterior cingulate gyrus total thickness	3.17 ± 0.28	3.34 ± 0.19	0.001	−4.097	
Right posterior cingulate gyrus thickness	3.14 ± 0.27	3.34 ± 0.20	0.001	−4.849	
Left posterior cingulate gyrus thickness	3.19 ± 0.31	3.34 ± 0.19	0.001	−3.270	
Parahippocampal gyrus total thickness	2.90 ± 0.22	2.95 ± 0.23	0.06 (*p* > 0.05)		
Right parahippocampal gyrus thickness	2.92 ± 0.26	3.00 ± 0.25	0.016		−2.407
Left parahippocampal gyrus thickness	2.86 ± 0.23	2.91 ± 0.26	0.18 (*p* > 0.05)		

## Data Availability

The data that support the findings of this study are available from the corresponding author upon reasonable request.

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
