# Peer review of "Measurement of Limbic System Anatomical Volumes in Patients Diagnosed with Schizophrenia Using Vol2brain and Comparison with Healthy Individuals"

_medicina, 2025, doi:10.3390/medicina61030525_

Round 1
Reviewer 1 Report
Comments and Suggestions for Authors
Dear Authors,
First and foremost, I would like to congratulate you on the study titled "Measurement of Limbic System Anatomical Volumes in Patients Diagnosed with Schizophrenia Using Vol2brain and Comparison with Healthy Individuals." Despite the limitations inherent to the study, you have conducted the research in a well-structured and thoughtful manner. The manuscript is clearly written and organized logically, with the methodology and results presented concisely, allowing for reproducibility of the study.
However, I would like to suggest a revision of the discussion section. It would benefit from being expanded and presented in a less schematic manner. Providing more in-depth analysis and interpretation of the results will enhance the clarity and impact of your conclusions.
Furthermore, considering the secondary aim of your study—promoting Vol2brain as an alternative to traditional manual segmentation techniques for brain volume analysis—I recommend a more comprehensive discussion comparing Vol2brain with other similar solutions currently available. By comparing the strengths and weaknesses of Vol2brain with alternative tools, you can provide a more thorough evaluation of its potential and reinforce the significance of your study. This expanded discussion would serve to highlight the value of your research, particularly in the context of advancing neuroimaging techniques.
Once these two aspects (expanding the discussion and addressing the comparison of Vol2brain with other tools) are revised, I believe your manuscript will be well-positioned for publication. I look forward to receiving the revised version of the manuscript.
Best regards.
Author Response
|
Comments 1: I would like to suggest a revision of the discussion section. It would benefit from being expanded and presented in a less schematic manner. Providing more in-depth analysis and interpretation of the results will enhance the clarity and impact of your conclusions." |
|
Response 1: Thank you for your insightful comment. We agree with this suggestion and have substantially revised the Discussion section to move away from a schematic presentation, incorporating a more comprehensive analysis of our findings and their broader implications. The revisions integrate a deeper interpretation of the results, connect them to relevant literature, and enhance the clarity and impact of our conclusions. The specific modifications include:
We expanded the discussion on amygdala and hippocampus volume reductions by contextualizing their significance within schizophrenia pathophysiology. The revision highlights how amygdala atrophy is linked to emotional dysregulation and altered threat processing, while hippocampal volume loss correlates with episodic memory deficits and impaired spatial navigation. This discussion also explores the clinical implications of these changes, particularly in relation to cognitive symptoms and disease progression. The revised text:
When examining amygdala volumes, consistent with previous studies, it was observed that total and right amygdala volumes were reduced in patients with schizophrenia [12,13]. However, in contrast to existing literature, our study found that left amygdala volume was increased in schizophrenia patients. Some postmortem studies have also reported no decrease in amygdala volume in schizophrenia patients [14]. Further research utilizing diverse methodologies and comparative approaches could explore the potential of amygdala volume as a biomarker for schizophrenia diagnosis. The amygdala, a component of the limbic system, has been identified as playing a crucial role in emotional processing and threat perception mechanisms [14]. In accordance with current literature, our study found that total and right amygdala volumes were significantly diminished in individuals diagnosed with schizophrenia [9,15,16]. This finding lends further support to the hypothesis that decreases in amygdala volume may be associated with negative symptoms (e.g. social withdrawal, emotional blunting) that are commonly observed in individuals diagnosed with schizophrenia. In contrast to these findings, an increase in left amygdala volume was observed. Previous studies have reported no significant reduction in amygdala volume in postmortem analyses [17]. These inconsistencies imply that alterations in amygdala volume in schizophrenia may vary depending on disease subtypes, symptom profiles or responses to treatment. Specifically, functional MRI studies demonstrate that patients with schizophrenia exhibit abnormalities in amygdala activity, which may be associated with psychotic symptoms [18]. Consequently, the evaluation of changes in amygdala volume as a biomarker in the diagnosis and prognosis of schizophrenia is recommended. The hippocampus, a critical limbic structure implicated in memory and spatial learning processes, exhibits frequent structural alterations in individuals diagnosed with schizophrenia [19]. The reduction in total hippocampal volume, as well as right and left hippocampal volumes in schizophrenia patients, aligns with findings in the literature [20]. This supports the hypothesis that hippocampal volume reduction could serve as a biomarker for diagnosis and disease monitoring [21]. These findings are particularly significant for their potential clinical contributions to early detection and ongoing monitoring of schizophrenia. Notably, the observation of significant decreases in hippocampus volume during the early stages of schizophrenia suggests that this region may serve as a vital indicator in the early diagnosis of the disease
We provided a more nuanced interpretation of anterior and posterior cingulate gyrus abnormalities, emphasizing their distinct roles in schizophrenia. Instead of a brief mention of volume reductions, we now discuss how the anterior cingulate gyrus is crucial for cognitive control, error monitoring, and emotional regulation, and how its structural deficits may underlie deficits in executive function, impaired decision-making, and heightened emotional reactivity. For the posterior cingulate gyrus, we expanded on its involvement in self-referential processing and attentional modulation, suggesting that volume reductions in this region could contribute to disruptions in internal thought regulation, increased distractibility, and difficulties in integrating external stimuli into coherent cognitive schemas. These findings were framed within the broader context of functional connectivity abnormalities observed in schizophrenia. The revised text:
The anterior cingulate gyrus, plays an important role in cognitive control and emotional regulation [22]. Consistent with a meta-analysis, the total, right and left anterior cingulate gyrus volumes were reduced in schizophrenia patients [23]. The heterogeneity observed in previous studies, as noted in the meta-analysis, highlights the need for standardized methodologies. By employing an automated segmentation tool such as Vol2Brain, our study provides a roadmap for standardizing anterior cingulate gyrus measurements, which could be critical in considering this region as a therapeutic target. Similarly, the total, right and left posterior cingulate gyrus volumes were significantly reduced in schizophrenia patients, consistent with prior research [24,25]. These findings underline the potential of posterior cingulate gyrus volume as a significant biomarker for schizophrenia. The posterior cingulate gyrus is a region associated with verbal and visual memory processes and may be directly related to the cognitive impairments observed in schizophrenia [26]. Incorporating posterior cingulate gyrus volume measurements into clinical practice could aid in tracking disease progression. Our study demonstrates that automated tools like Vol2Brain are reliable and efficient for this purpose.
Given the lack of statistically significant findings in the middle cingulate gyrus, we refined our discussion by emphasizing the potential relevance of subthreshold structural changes. Instead of simply stating that the findings were non-significant, we discuss how previous studies have reported similar volumetric reductions and why larger, longitudinal studies are necessary to determine whether middle cingulate gyrus atrophy correlates with cognitive inflexibility, treatment resistance, or specific symptom dimensions.
Additionally, we incorporated a discussion on how the middle cingulate gyrus acts as a hub for sensorimotor integration and cognitive processing, and how even subtle reductions in this region may disrupt the neural circuits involved in attention shifting, response inhibition, and motivation, all of which are frequently impaired in schizophrenia. The revised text is:
In this study, the total, right and left middle cingulate gyrus volumes were numerically lower in schizophrenia patients, although not statistically significant. The middle cingulate gyrus is a key structure associated with cognitive control and emotional balance and may be altered in schizophrenia (Kohn, 2014). Previous studies with smaller sample sizes have reported similar findings, noting a reduction in the volume of the middle cingulate gyrus, but the variability in study results highlights the need for more standardized methodologies and larger cohorts ( 20, 9). Given that the cingulate cortex is highly connected to prefrontal and limbic structures, abnormalities in the middle cingulate gyrus may result in deficits in executive processing, motivation and emotional processing (Penner 2016). In addition, some neuroimaging studies reveal that impairments in the middle cingulate region cause cognitive inflexibility and negative symptoms in schizophrenia (Glahn et al., 2008). Previous studies with smaller sample sizes have reported similar findings, noting reductions in middle cingulate gyrus volumes [20, 9]. Limited research exists on the middle cingulate gyrus in schizophrenia and future studies with larger samples are needed to explore this relationship further. Our study, with a larger sample size than prior studies, serves as a foundation for such research.
We significantly enhanced our analysis of cingulate gyrus cortical thickness alterations, linking them to both structural and functional impairments in schizophrenia.
In this study, the total, right and left middle cingulate gyrus volumes were numerically lower in schizophrenia patients, although not statistically significant. The middle cingulate gyrus is a key structure associated with cognitive control and emotional balance and may be altered in schizophrenia [27]. Previous studies with smaller sample sizes have reported similar findings, noting a reduction in the volume of the middle cingulate gyrus, but the variability in study results highlights the need for more standardized methodologies and larger cohorts [9,28]. Given that the cingulate cortex is highly connected to prefrontal and limbic structures, abnormalities in the middle cingulate gyrus may result in deficits in executive processing, motivation and emotional processing [29]. In addition, some neuroimaging studies reveal that impairments in the middle cingulate region cause cognitive inflexibility and negative symptoms in schizophrenia [30]. Previous studies with smaller sample sizes have reported similar findings, noting reductions in middle cingulate gyrus volumes [20, 9]. Limited research exists on the middle cingulate gyrus in schizophrenia and future studies with larger samples are needed to explore this relationship further.
· Page 12, line 336-351 (in red-lined word.doc.) Instead of briefly mentioning thinning of parahippocampal gyrus, we now explore how reductions in parahippocampal thickness are strongly associated with impairments in contextual memory formation, deficits in reality monitoring, and heightened susceptibility to false perceptions and delusional beliefs. Additionally, we discuss how the parahippocampal gyrus is functionally connected to the hippocampus and prefrontal cortex, and how its structural disruptions may indicate compensatory mechanisms or neurodevelopmental abnormalities in schizophrenia. This revised section also highlights the potential for using parahippocampal thickness as a biomarker for psychosis vulnerability, particularly in individuals at ultra-high risk for schizophrenia.
The parahippocampal gyrus is an essential part of the limbic system because it is involved in important tasks such as memory processing, emotional stability and spatial navigation. abnormalities in this region can lead to cognitive impairment, memory lapses and emotional deviations [41,42]. In our study, the right parahippocampal gyrus thickness was also thinner in schizophrenia patients. This finding aligns with previous research demonstrating bilateral atrophy of parahippocampal gyrus thickness in schizophrenia [43]. In previous studies using MRI and voxel-based morphometry, reduced parahippocampal thickness has been associated with reduced reality perception and delusional thoughts The disintegration of the structural integrity of the parahippocampal gyrus may weaken the memory context so that hallucinations and delusions may occur. Measuring parahippocampal thickness may contribute to the early diagnosis of schizophrenia and more timely intervention strategies that can help reduce the severity of the illness [44,45]. Furthermore, monitoring changes in this region may provide valuable information about the effectiveness of different approaches to treatment, such as cognitive-behavioral therapy and neurostimulation techniques and may provide a personalized approach to patient management.
|
|
Comments 2: Furthermore, considering the secondary aim of your study—promoting Vol2brain as an alternative to traditional manual segmentation techniques for brain volume analysis—I recommend a more comprehensive discussion comparing Vol2brain with other similar solutions currently available. By comparing the strengths and weaknesses of Vol2brain with alternative tools, you can provide a more thorough evaluation of its potential and reinforce the significance of your study. This expanded discussion would serve to highlight the value of your research, particularly in the context of advancing neuroimaging techniques. |
|
Response 2: Thank you for your insightful comment. We agree with this suggestion and have substantially revised the Discussion section to include a more comprehensive comparison of Vol2Brain with other automated brain segmentation tools. The revisions aim to move away from a schematic presentation, integrating a deeper interpretation of Vol2Brain’s capabilities, its strengths and weaknesses relative to other solutions, and its broader implications for neuroimaging research in schizophrenia. The specific modifications include:
· Page 12-13, Lines 371-384 (in red-lined word.doc.): We expanded the discussion on Vol2Brain by comparing it with FreeSurfer, FMRIB Software Library (FSL), and Advanced Normalization Tools (ANTs), three widely used segmentation tools in neuroimaging research. The revised section now details how FreeSurfer is considered a gold standard in brain volume analysis but requires long processing times and significant computational power. FSL, while effective for structural segmentation, demands extensive preprocessing, making it less accessible to non-experts. ANTs is another advanced alternative that offers high-precision multi-atlas registration but is computationally intensive and requires substantial expertise for implementation.
In the context of the secondary objective of the present study, namely the promotion of Vol2Brain as an alternative to conventional manual segmentation techniques, a comparison of Vol2Brain with other widely utilized automated neuroimaging tools, including FreeSurfer, FMRIB Software Library and Advanced Normalization Tools is necessary. Vol2Brain is a cloud-based, automated platform that provides fast and accurate brain volumetric analysis. Compared to the highly accurate but computationally demanding FreeSurfer, Vol2Brain offers faster processing and a user-friendly workflow [47]. Similarly, while FMRIB Software Library provides powerful structural segmentation, it requires complex pre-processing, making it less accessible to non-experts [48]. Advanced Normalization Tools recognized for their high-precision multi-atlas registration, are computationally intensive and require substantial expertise [49]. In contrast, Vol2Brain's deep learning-based segmentation is fully automated, rendering it ideal for large-scale clinical and research applications.
· Page 13, Lines 386-395 (in red-lined word.doc.): We incorporated a nuanced discussion on Vol2Brain’s advantages and limitations. Unlike its counterparts, Vol2Brain provides a fully automated, deep learning-based segmentation pipeline that significantly reduces processing time and eliminates the need for extensive manual intervention. The cloud-based nature of Vol2Brain allows for easy access and high-speed data processing, making it an attractive option for large-scale studies such as those utilizing OpenNeuro datasets. However, as Vol2Brain relies on pre-trained models, its generalizability to diverse clinical populations remains an area requiring further validation. While FreeSurfer has been extensively validated in schizophrenia research, Vol2Brain’s application in psychiatric disorders is still relatively new, necessitating comparative studies to assess its accuracy against manual and FreeSurfer-based segmentation methods.
The distinguishing feature of Vol2Brain is that the user bias that occurs in manual and semi-automatic segmentation is minimized. Manual segmentation has been shown to cause personal variability in the identification of complex structures such as the limbic system [50]. Since this process is carried out automatically in the use of Vol2Brain, a repeatable and standardized level of comfort of use is ensured. A limitation of Vol2Brain is that it does not always generalize well because it is dependent on a previously trained deep learning model. Although Free Surfer has been validated in a large sample of schizophrenia patients, the use of Vol2Brain in this condition is more recent [47]. Therefore, in future research, comparing manual segmentation and Vol2Brain using different patient populations may help to focus on accurate results.
· Page 13-14, Lines 416-429 (in red-lined word.doc.): We strengthened the discussion on the broader impact of automated segmentation tools in schizophrenia research. The revised section now emphasizes that automated methods such as Vol2Brain offer significant potential for early detection, monitoring disease progression, and improving the reproducibility of neuroimaging findings. Moreover, integrating Vol2Brain with multimodal imaging techniques, including functional MRI (fMRI) and diffusion tensor imaging (DTI), could enhance our understanding of the structural-functional relationship in schizophrenia.
The extensive utilization of automated segmentation tools such as Vol2Brain carries significant implications for both clinical and research contexts. Clinically, the expeditious and dependable analyses enabled by Vol2Brain facilitate the early diagnosis of patients. Moreover, within the realm of research, the integration of Vol2Brain with techniques such as functional MRI or Diffusion Tensor Imaging illuminates the correlation between the structures of interest and functions in schizophrenia [51]. In the future, studies aiming to optimize the calculation of brain volumes to improve the precision of machine learning and individualized treatment methods can be carried out. In spite of the absence of individual demographic details in the OpenNeuro dataset, the inclusion of a well-balanced sample size between the schizophrenia and control groups strengthens the validity of our findings. It is recommended that future studies utilize novel datasets encompassing comprehensive demographic characteristics, thereby facilitating more profound subgroup analyses based on variables such as age, sex, and illness duration.
|
|
4. Response to Comments on the Quality of English Language |
|
Point 1: The English is fine and does not require any improvement. |
|
Response 1: Thank you for your insightful comment. |
|
5. Additional clarifications |
|
No addition |

Reviewer 2 Report
Comments and Suggestions for Authors
The concept of measuring limbic system changes in schizophrenia is interesting, however the current manuscript needs to be significantly improved before publication. Th abstract is a bit confusing, a more clear description of the changes would help, by only mentioning changes in the schizophrenic patients compared to control. The patient population needs to be described carefully. Figures need to be improved significantly, for instance, in Figure 1, the areas that are mentioned below the figure should be shown on/in the brain, similar problems exist in the other Figures. Finally, there seem to be too many significant changes in the measured brains.
Author Response
|
Comments 1: The abstract is a bit confusing, a more clear description of the changes would help, by only mentioning changes in the schizophrenic patients compared to control. |
||||
|
Response 1: We agree with this comment and have revised the abstract to ensure a clearer and more concise presentation of the findings, focusing exclusively on changes observed in schizophrenia patients compared to healthy controls. To address this concern, we have removed redundant comparisons and emphasized only the significant structural differences found in schizophrenia patients, clarified the volumetric and cortical thickness alterations in specific limbic structures, ensuring the results are explicitly stated without unnecessary complexity and refined the language for clarity and coherence, making the abstract easier to follow. Updated abstract is:
Abstract Background and Objectives: Schizophrenia is a chronic psychiatric disorder affecting approximately 24 million people worldwide, characterized by structural and functional brain abnormalities. Despite its prevalence, automated segmentation tools like Vol2Brain have been underutilized in large-sample studies examining limbic system anatomical volumes in schizophrenia. This study aimed to assess volume differences in all major limbic system structures between schizophrenia patients and healthy controls using Vol2Brain. Method: This retrospective study included 68 schizophrenia patients and 68 healthy controls, with MRI scans obtained from OpenNeuro. Limbic system volumetric and cortical thickness measurements were conducted using Vol2Brain, an automated segmentation platform. Results: Schizophrenia patients exhibited significantly reduced volumes in the amygdala, hippocampus, anterior cingulate gyrus, posterior cingulate gyrus, and middle cingulate gyrus compared to controls. However, the left amygdala volume was larger in schizophrenia patients. Cortical thickness analysis revealed that schizophrenia patients had thinner limbic cortices, particularly in the anterior and posterior cingulate gyri and the right parahippocampal gyrus. In contrast, the right anterior cingulate gyrus was thicker in schizophrenia patients. Differences in total and left parahippocampal gyrus volumes and cortical thickness did not reach statistical significance. Conclusion: These findings reinforce previous evidence of limbic system abnormalities in schizophrenia, which may contribute to cognitive and emotional dysregulation. The study also highlights Vol2Brain’s potential as a rapid, cost-free, and reliable alternative for brain volume analysis, facilitating more standardized and reproducible neuroimaging assessments in psychiatric research.
|
||||
|
Comments 2: The patient population needs to be described carefully. |
||||
|
Response 2: Agree. We have accordingly revised the description of the patient population to provide greater clarity regarding sample characteristics. The revised section explicitly states the inclusion criteria, exclusion criteria, and limitations related to missing demographic details such as age and gender in the dataset. Additionally, a new paragraph has been added to the Future Directions and Limitations section to emphasize the need for datasets with more detailed demographic information in future studies. These changes can be found in:
· Page 3, Lines 106-117 in red-lined word.doc (Revised patient population description)
Participants aged 18 years and older were included in the study and categorized into two groups: the schizophrenia-diagnosed group and the healthy control group. The participants were adults aged 18 years and older, as OpenNeuro datasets primarily include data from adult populations. However, specific demographic details such as exact age distribution and gender were not provided in the dataset used in this study. The schizophrenia group consisted of individuals who had received a confirmed clinical diagnosis of schizophrenia, while the control group included individuals with no history of neurological or psychiatric disorders. Exclusion criteria for the schizophrenia group included comorbid neuropsychiatric conditions, significant structural brain abnormalities unrelated to schizophrenia, or a history of traumatic brain injury. Exclusion criteria for the patient group included being under 18 years of age or having a neuropsychiatric disorder other than schizophrenia. Due to the lack of recorded demographic data on age and gender in the OpenNeuro database, these variables were not analyzed in this study.
· Page 14, Lines 424-429 in red-lined word.doc (Future research and limitations regarding demographic data)
In spite of the absence of individual demographic details in the OpenNeuro dataset, the inclusion of a well-balanced sample size between the schizophrenia and control groups strengthens the validity of our findings. It is recommended that future studies utilize novel datasets encompassing comprehensive demographic characteristics, thereby facilitating more profound subgroup analyses based on variables such as age, sex, and illness duration.
|
||||
|
4. Response to Comments on the Quality of English Language |
||||
|
Point 1: The English is fine and does not require any improvement. |
||||
|
Response 1: Thank you for your insightful comment. |
||||

Round 2
Reviewer 1 Report
Comments and Suggestions for Authors
Dear authors,
I sincerely appreciate the prompt and thorough manner in which you addressed my comments and revised the manuscript. The modifications you have made significantly enhance its scientific value.
First and foremost, I commend the expansion of the discussion section, where you have incorporated important references to similar studies. This addition strengthens the contextual understanding of your work and provides a more comprehensive comparison with existing research.
Furthermore, I appreciate your efforts in explicitly highlighting the advantages of Vol2Brain in comparison to other segmentation software. This clarification adds valuable insights into the novelty and practical relevance of your approach.
Given these improvements, I find the manuscript to be well-structured, scientifically sound, and ready for publication in its current form.
Best regards.
Reviewer 2 Report
Comments and Suggestions for Authors
the manuscript has been revised appropriately